# Rate-Agnostic (Causal) Structure Learning

**Sergey Plis**
The Mind Research Network,
Albuquerque, NM
s.m.plis@gmail.com

**David Danks**
Carnegie-Mellon University
Pittsburgh, PA
ddanks@cmu.edu

**Cynthia Freeman**
The Mind Research Network,
CS Dept., University of New Mexico
Albuquerque, NM
cynthiaw2004@gmail.com

**Vince Calhoun**
The Mind Research Network
ECE Dept., University of New Mexico
Albuquerque, NM
vcalhoun@mrn.org

## Abstract

Causal structure learning from time series data is a major scientific challenge. Extant algorithms assume that measurements occur sufficiently quickly; more precisely, they assume approximately equal system and measurement timescales. In many domains, however, measurements occur at a significantly slower rate than the underlying system changes, but the size of the timescale mismatch is often unknown. This paper develops three causal structure learning algorithms, each of which discovers all dynamic causal graphs that explain the observed measurement data, perhaps given undersampling. That is, these algorithms all learn causal structure in a "rate-agnostic" manner: they do not assume any particular relation between the measurement and system timescales. We apply these algorithms to data from simulations to gain insight into the challenge of undersampling.

## 1 Introduction

Dynamic causal systems are a major focus of scientific investigation in diverse domains, including neuroscience, economics, meteorology, and education. One significant limitation in all of these sciences is the difficulty of measuring the relevant variables at an appropriate timescale for the particular scientific domain. This challenge is particularly salient in neuroimaging: standard fMRI experiments sample the brain's bloodflow approximately every one or two seconds, though the underlying neural activity (i.e., the major driver of bloodflow) occurs much more rapidly. Moreover, the precise timescale of the underlying causal system is unknown; it is almost certainly faster than the fMRI measurements, but it is unknown how *much* faster.

In this paper, we aim to learn the causal structure of a system that evolves at timescale $\tau_S$, given measurements at timescale $\tau_M$. We focus on the case in which $\tau_S$ is faster than $\tau_M$ to an unknown degree. We assume that the underlying causal structure can be modeled as a directed graphical model $\mathbf{G}$ without simultaneous influence. There has been substantial work on modeling the statistics of time series, but relatively less on learning causal structure, and almost all of that assumes that the measurement and causal timescales match [1–5]. The problem of causal learning from "undersampled" time series data was explicitly addressed by [6, 7], but they assumed that the degree of undersampling—i.e., the ratio of $\tau_S$ to $\tau_M$—was both known and small. In contrast, we focus on the significantly harder challenge of causal learning when that ratio is unknown.

We provide a formal specification of the problem and representational framework in Section 2. We then present three different Rate-Agnostic Structure Learning (RASL) algorithms in Section 3. We finish in Section 4 by exploring their performance on synthetic data.

## 2 Representation and Formalism

A dynamic causal graphical model consists of a graph $\mathbf{G}$ over random variables $\mathbf{V}$ at the current time $t$, as well as nodes for $\mathbf{V}$ at all previous (relative) timesteps that contain a direct cause of a variable at the current timestep.[1] The *Markov order* of the system is the largest $k$ such that $V_i^{t-k} \rightarrow V_j^t$, where superscripts denote timestep. We assume throughout that the "true" underlying causal system is Markov order 1, and that all causally relevant variables are measured.[2] Finally, we assume that there are no isochronal causal edges $V_i^t \rightarrow V_j^t$; causal influences inevitably take time to propagate, and so any apparent isochronal edge will disappear when measured sufficiently finely. Since we do not assume that the causal timescale $\tau_S$ is known, this is a relatively innocuous assumption.

$\mathbf{G}$ is thus over $2\mathbf{V}$ nodes, where the only edges are $V_i^{t-1} \rightarrow V_j^t$, where possibly $i = j$. There is additionally a conditional probability distribution or density, $P(\mathbf{V}^t|\mathbf{V}^{t-1})$, which we assume to be time-independent. We do *not*, however, assume stationarity of $P(\mathbf{V}^t)$. Finally, we assume appropriate versions of the Markov ("Variable $V$ is independent of non-descendants given parents") and Faithfulness/Stability ("The only independencies are those implied by Markov") assumptions, such that the graph and probability distribution/density mutually constrain each other.

Let $\{t^0, t^1, \ldots, t^k, \ldots\}$ be the measurement timesteps. We *undersample at rate* $u$ when we measure only timesteps $\{t^0, t^u, \ldots, t^{uk}, \ldots\}$; the causal timescale is thus "undersampled at rate 1." We denote the causal graph resulting from undersampling at rate $u$ by $\mathbf{G}^u$. To obtain $\mathbf{G}^u$, we "unroll" $\mathbf{G}^1$ by introducing nodes for $\mathbf{V}^{t-2}$ that bear the same graphical and parametric relations to $\mathbf{V}^{t-1}$ as those variables bear to $\mathbf{V}^t$, and iterate until we have included $\mathbf{V}^{t-u}$. We then marginalize out all variables except those in $\mathbf{V}^t$ and $\mathbf{V}^{t-u}$.

Marginalization yields an Acyclic Directed Mixed Graph (ADMG) $\mathbf{G}^u$ containing both directed and bidirected edges [8]. $V_i^{t-u} \rightarrow V_j^t$ in $\mathbf{G}^u$ iff there is a directed path from $V_i^{t-u}$ to $V_j^t$ in the unrolled graph. Define a *trek* to be a pair of directed paths $(\pi_1, \pi_2)$ such that both have the same start variable. $V_i^t \leftrightarrow V_j^t$ in $\mathbf{G}^u$ iff there is a trek between $V_i^t$ and $V_j^t$ with $length(\pi_1) = length(\pi_2) = k < u$. Clearly, if a bidirected edge occurs in $\mathbf{G}^m$, then it occurs in $\mathbf{G}^u$ for all $u \geq m$.

Unrolling-and-marginalizing can be computationally complex due to duplication of nodes, and so we instead use *compressed graphs* that encode temporal relations in edges. For an arbitrary dynamic causal graph $\mathbf{H}$, $\mathcal{H}$ is its compressed graph representation: (i) $\mathcal{H}$ is over non-time-indexed nodes for $\mathbf{V}$; (ii) $V_i \rightarrow V_j$ in $\mathcal{H}$ iff $V_i^{t-1} \rightarrow V_j^t$ in $\mathbf{H}$; and (iii) $V_i \leftrightarrow V_j$ in $\mathcal{H}$ iff $V_i^t \leftrightarrow V_j^t$ in $\mathbf{H}$. Compressed graphs can be cyclic ($V_i \rightleftarrows V_j$ for $V_i^{t-1} \rightarrow V_j^t$ and $V_j^{t-1} \rightarrow V_i^t$), including self-cycles. There is clearly a 1-1 mapping between dynamic ADMGs and compressed graphs.

Computationally, the effects of undersampling at rate $u$ can be computed in a compressed graph simply by finding directed paths of length $u$ in $\mathcal{G}^1$. More precisely, $V_i^{t-u} \rightarrow V_j^t$ in $\mathcal{G}^u$ iff there is a directed path of length $u$ in $\mathcal{G}^1$. Similarly, $V_i^t \leftrightarrow V_j^t$ in $\mathcal{G}^u$ iff there is a trek with $length(\pi_1) = length(\pi_2) = k < u$ in $\mathcal{G}^1$. We thus use compressed graphs going forward.

## 3 Algorithms

The core question of this paper is: given $\mathcal{H} = \mathcal{G}^u$ for unknown $u$, what can be inferred about $\mathcal{G}^1$? Let $[\![\mathcal{H}]\!] = \{\mathcal{G}^1 : \exists u\, \mathcal{G}^u = \mathcal{H}\}$ be the equivalence class of $\mathcal{G}^1$ that could, for some undersample rate, yield $\mathcal{H}$. We are thus trying to learn $[\![\mathcal{H}]\!]$ from $\mathcal{H}$. An obvious brute-force algorithm is: for each possible $\mathcal{G}^1$, compute the corresponding graphs for all $u$, and then output all $\mathcal{G}^u = \mathcal{H}$. Equally obviously, this algorithm will be computationally intractable for any reasonable $n$, as there are $2^{n^2}$ possible $\mathcal{G}^1$ and $u$ can (in theory) be arbitrarily large. Instead, we pursue three different constructive strategies that more efficiently "build" the members of $[\![\mathcal{H}]\!]$ (Sections 3.2, 3.3, and 3.4). Because these algorithms make no assumptions about $u$, we refer to them each as *RASL*—Rate Agnostic Structure Learner—and use subscripts to distinguish between different types. First, though, we provide some key theoretical results about *forward* inference that will be used by all three algorithms.

## 3.1 Nonparametric Forward Inference

For given $\mathcal{G}^1$ and $u$, there is an efficient algorithm [9] for calculating $\mathcal{G}^u$, but it is only useful in learning if we have stopping rules that constrain which $\mathcal{G}^1$ and $u$ should ever be considered. These rules will depend on how $\mathcal{G}^1$ changes as $u \to \infty$. A key notion is a *strongly connected component* (SCC) in $\mathcal{G}^1$: a maximal set of variables $\mathbf{S} \subseteq \mathbf{V}$ such that, for every $X, Y \in \mathbf{S}$ (possibly $X = Y$), there is a directed path from $X$ to $Y$. Non-singleton SCCs are clearly cyclic and can provably be decomposed into a set of (possibly overlapping) simple loops (i.e., those in which no node is repeated): $\sigma_1, \ldots, \sigma_s$ [10]. Let $\mathcal{L}_\mathbf{S}$ be the set of those simple loop lengths.

One stopping rule must specify, for given $\mathcal{G}^1$, which $u$ to consider. For a single SCC, the greatest common divisor of simple loop lengths (where $\gcd(\mathcal{L}_\mathbf{S}) = 1$ for singleton $\mathbf{S}$) is key: $\gcd(\mathcal{L}_\mathbf{S}) = 1$ iff $\exists f$ s.t. $\forall u > f[\mathcal{G}^u = \mathcal{G}^f]$; that is, $\gcd()$ determines whether an SCC "converges" to a fixed-point graph as $u \to \infty$. We can constrain $u$ if there is such a fixed-point graph, and Theorem 3.1 generalizes [9, Theorem 5] to provide an upper bound on (interesting) $u$. (All proofs found in supplement.)

**Theorem 3.1.** *If* $\gcd(\mathcal{L}_\mathbf{S}) = 1$, *then stabilization occurs at* $f \leq n_F + \gamma + d + 1$.

where $n_F$ is the Frobenius number,[3] $d$ is the graph diameter, and $\gamma$ is the transit number (see supplement). This is a theoretically useful bound, but is not practically helpful since neither $\gamma$ nor $n_F$ have a known analytic expression. Moreover, $\gcd(\mathcal{L}_\mathbf{S}) = 1$ is a weak restriction, but a restriction nonetheless. We instead use a functional stopping rule for $u$ (Theorem 3.2) that holds for all $\mathcal{G}$:

**Theorem 3.2.** *If* $\mathcal{G}^u = \mathcal{G}^v$ *for* $u > v$, *then* $\forall w > u \exists k_w < u[\mathcal{G}^w = \mathcal{G}^{k_w}]$.

That is, as $u$ increases, if we find a graph that we previously encountered, then there cannot be any new graphs as $u \to \infty$. For a given $\mathcal{G}^1$, we can thus determine *all* possible corresponding undersampled graphs by computing $\mathcal{G}^2, \mathcal{G}^3, \ldots$ until we encounter a previously-observed graph. This stopping rule enables us to (correctly) constrain the $u$ that are considered for each $\mathcal{G}^1$.

We also require a stopping rule for $\mathcal{G}^1$, as we cannot evaluate all $2^{n^2}$ possible graphs for any reasonable $n$. The key theoretical result is:

**Theorem 3.3.** *If* $\mathcal{G}^1 \subseteq \mathcal{J}^1$, *then* $\forall u[\mathcal{G}^u \subseteq \mathcal{J}^u]$.

Let $\mathcal{G}^1_\mathbf{E}$ be the graph resulting from adding the edges in $\mathbf{E}$ to $\mathcal{G}^1$. Since this is simply another graph, it can be undersampled at rate $u$; denote the result $(\mathcal{G}^1_\mathbf{E})^u$. Since $\mathcal{G}^1_\mathbf{E}$ can always serve as $\mathcal{J}^1$ in Theorem 3.3, we immediately have the following two corollaries:

**Corollary 3.4.** *If* $\mathcal{G}^u \nsubseteq \mathcal{H}$, *then* $\forall \mathbf{E}[(\mathcal{G}^1_\mathbf{E})^u \nsubseteq \mathcal{H}]$

**Corollary 3.5.** *If* $\forall u[\mathcal{G}^u \nsubseteq \mathcal{H}]$, *then* $\forall \mathbf{E}, u[(\mathcal{G}^1_\mathbf{E})^u \nsubseteq \mathcal{H}]$

We thus have a stopping rule for some candidate $\mathcal{G}^1$: if $\mathcal{G}^u$ is not an edge-subset of $\mathcal{H}$ for all $u$, then do not consider any edge-superset of $\mathcal{G}^1$. This stopping rule fits very cleanly with "constructive" algorithms that iteratively add edge(s) to candidate $\mathcal{G}^1$. We now develop three such algorithms.

## 3.2 A recursive edgewise inverse algorithm

The two stopping rules naturally suggest a recursive structure learning algorithm with $\mathcal{H}$ as input and $[\![\mathcal{H}]\!]$ as output. Start with an empty graph. For each edge $e$ (of $n^2$ possible edges), construct $\mathcal{G}^1$ containing only $e$. If $\mathcal{G}^u \nsubseteq \mathcal{H}$ for all $u$, then reject; else if $\mathcal{G}^u = \mathcal{H}$ for some $u$,[4] then add $\mathcal{G}^1$ to $[\![\mathcal{H}]\!]$; else, recurse into non-conflicting graphs in order. Effectively, this is a depth first search (DFS) algorithm on the solution tree; denote it as *RASL_re* for "recursive edgewise." Figure 1a provides pseudo-code, and Figure 1b shows how one DFS path in the search tree unfolds. We can prove:

**Theorem 3.6.** *The RASL_re algorithm is correct and complete.*

One significant drawback of RASL_re is that the same graph can be constructed in many different ways, corresponding to different orders of edge addition; the search tree is actually a search lat-

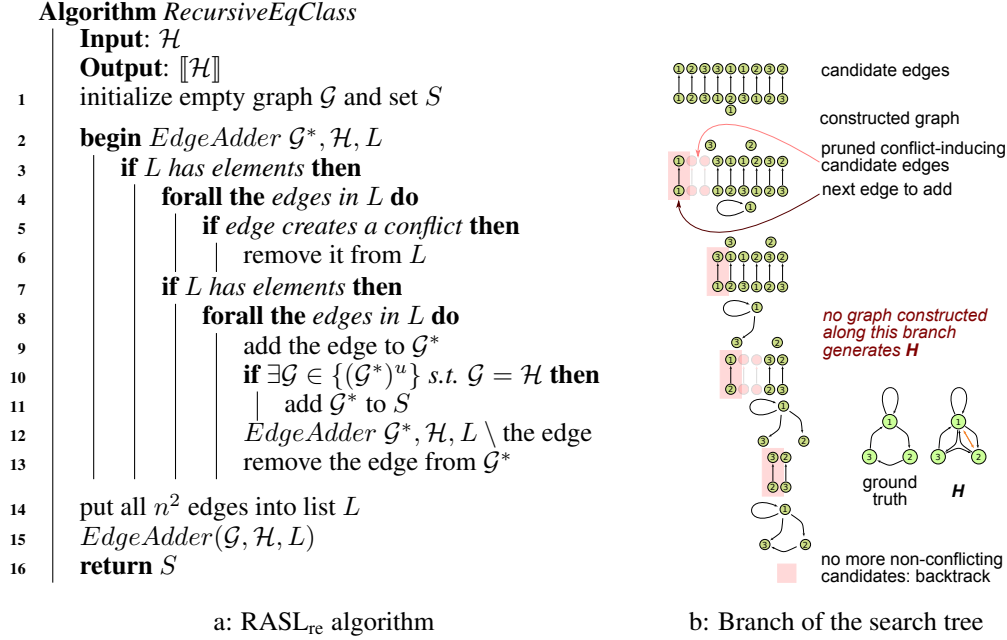

**Algorithm** *RecursiveEqClass*

> **Input**: $\mathcal{H}$
> **Output**: $[\![\mathcal{H}]\!]$

1    initialize empty graph $\mathcal{G}$ and set $S$

2    **begin** $EdgeAdder\ \mathcal{G}^*, \mathcal{H}, L$

3      **if** $L$ *has elements* **then**

4        **forall the** *edges in* $L$ **do**

5          **if** *edge creates a conflict* **then**

6            remove it from $L$

7      **if** $L$ *has elements* **then**

8        **forall the** *edges in* $L$ **do**

9          add the edge to $\mathcal{G}^*$

10          **if** $\exists \mathcal{G} \in \{(\mathcal{G}^*)^u\}$ *s.t.* $\mathcal{G} = \mathcal{H}$ **then**

11            add $\mathcal{G}^*$ to $S$

12          $EdgeAdder\ \mathcal{G}^*, \mathcal{H}, L \setminus$ the edge

13          remove the edge from $\mathcal{G}^*$

14    put all $n^2$ edges into list $L$

15    $EdgeAdder(\mathcal{G}, \mathcal{H}, L)$

16    **return** $S$

a: RASL$_{\text{re}}$ algorithm

b: Branch of the search tree

Figure 1: RASL$_{\text{re}}$ algorithm 1a specification, and 1b search tree example

tice. The algorithm is thus unnecessarily inefficient, even when we use dynamic programming via memoization of input graphs.

### 3.3 An iterative edgecentric inverse algorithm

To minimize multiple constructions of the same graph, we can use *RASL$_{ie}$* ("iterative edgewise") which generates, at stage $i$, all not-yet-eliminated $\mathcal{G}^1$ with exactly $i$ edges. More precisely, at stage 0, RASL$_{ie}$ starts with the empty graph; if $\mathcal{H}$ is also empty, then it adds the empty graph to $[\![\mathcal{H}]\!]$. Otherwise, it moves to stage 1. In general, at stage $i + 1$, RASL$_{ie}$ (a) considers each graph $\mathcal{G}^1$ resulting from a single edge addition to an acceptable graph at stage $i$; (b) rejects $\mathcal{G}^1$ if it conflicts (for all $u$) with $\mathcal{H}$; (c) otherwise keeps $\mathcal{G}^1$ as acceptable at $i + 1$; and (d) if $\exists u[\mathcal{G}^u = \mathcal{H}]$, then adds $\mathcal{G}^1$ to $[\![\mathcal{H}]\!]$. RASL$_{ie}$ continues until there are no more edges to add (or it reaches stage $n^2 + 1$). Figure 2 provides the main loop (Figure 2a) and core function of RASL$_{ie}$ (Figure 2c), as well as an example of the number of graphs potentially considered at each stage (Figure 2b). RASL$_{ie}$ provides significant speed-up and memory gains over RASL$_{re}$ (see Figure 3).

We optimize RASL$_{ie}$ by tracking the single edges that could possibly still be added; for example, if a single-edge graph is rejected in stage 1, then do not consider adding that edge at other stages. Additional conflicts can be derived analytically, further reducing the graphs to consider. In general, absence of an edge in $\mathcal{H}$ implies, for the corresponding (unknown) $u$, absence of length $u$ paths in all $\mathcal{G}^1 \in [\![\mathcal{H}]\!]$. Since we do not know $u$, we cannot directly apply this constraint. However, lemmas 3.7 and 3.8 provide useful, special case constraints for $u > 1$ (implied by a single bidirected edge).

**Lemma 3.7.** *If $u > 1$, then $\forall V \nrightarrow W \in \mathcal{H}$, $\mathcal{G}^1$ cannot contain any of the following paths: 1. $\overset{\curvearrowright}{V} \to W$; 2. $\overset{\curvearrowright}{V} \to X \to W$; 3. $V \to \overset{\curvearrowright}{X} \to W$; 4. $V \to X \to \overset{\curvearrowright}{W}$; 5. $V \to \overset{\curvearrowright}{W}$.*

**Lemma 3.8.** *If $u > 1$, then $\forall V \nleftrightarrow W \in \mathcal{H}$ $\nexists T[V \leftarrow T \to W] \in \mathcal{G}^1$*

### 3.4 An iterative loopcentric inverse algorithm

RASL$_{ie}$ yields results in reasonable time for $\mathcal{H}$ with up to 8 nodes, though it is computationally demanding. We can gain further computational advantages if we assume that $\mathcal{H}$ is an SCC. This assumption is relatively innocuous, as it requires only that our time series be generated by a system with (appropriate) feedback loops. As noted earlier, any SCC is composed of a set of simple loops, and so we modify RASL$_{ie}$ to iteratively add *loops* instead of edges; call the resulting algorithm

**Algorithm** *IterativeEqClass*
  **Input**: $\mathcal{H}$
  **Output**: $[\![\mathcal{H}]\!]$

1.   initialize empty sets $S$
2.   init $d$ as an empty graph and $n^2$ edges
3.   **while** $d$ **do**
4.     $d, S_i = NextIterationGraphs(d, \mathcal{H})$
5.     $S = S \cup S_i$
6.   **return** $S$

a: RASL$_{ie}$ main algorithm

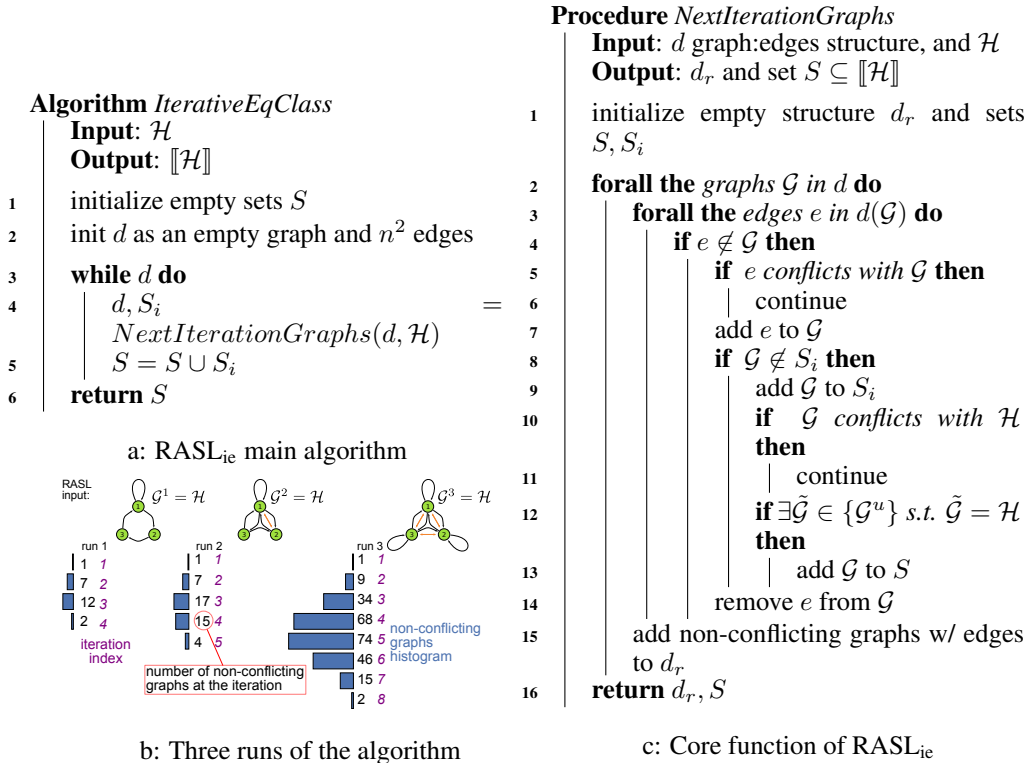

b: Three runs of the algorithm

**Procedure** *NextIterationGraphs*
  **Input**: $d$ graph:edges structure, and $\mathcal{H}$
  **Output**: $d_r$ and set $S \subseteq [\![\mathcal{H}]\!]$

1.   initialize empty structure $d_r$ and sets $S, S_i$
2.   **forall the** *graphs* $\mathcal{G}$ in $d$ **do**
3.     **forall the** *edges* $e$ in $d(\mathcal{G})$ **do**
4.       **if** $e \notin \mathcal{G}$ **then**
5.         **if** $e$ *conflicts with* $\mathcal{G}$ **then**
6.           continue
7.         add $e$ to $\mathcal{G}$
8.         **if** $\mathcal{G} \notin S_i$ **then**
9.           add $\mathcal{G}$ to $S_i$
10.           **if** $\mathcal{G}$ *conflicts with* $\mathcal{H}$ **then**
11.             continue
12.           **if** $\exists \tilde{\mathcal{G}} \in \{\mathcal{G}^u\}$ *s.t.* $\tilde{\mathcal{G}} = \mathcal{H}$ **then**
13.             add $\mathcal{G}$ to $S$
14.         remove $e$ from $\mathcal{G}$
15.   add non-conflicting graphs w/ edges to $d_r$
16.   **return** $d_r, S$

c: Core function of RASL$_{ie}$

Figure 2: RASL$_{ie}$ algorithm (a) main loop; (b) example of graphs considered; and (c) core function.

RASL$_{il}$ for "iterative loopwise." More precisely, RASL$_{il}$ uses the same algorithm as in Figure 2, but successively attempts to add non-conflicting simple *loops*, rather than non-conflicting edges. RASL$_{il}$ also incorporates the additional constraints due to lemmas 3.7 and 3.8.

RASL$_{il}$ is surprisingly much faster than RASL$_{ie}$ even though, for $n$ nodes, there are $\sum_{i=0}^{n} \binom{n}{i}(i-1)!$ simple loops (compared to $n^2$ edges). The key is that introducing a single simple loop induces multiple constraints simultaneously, and so conflicting graphs are discovered at a much earlier stage. As a result, RASL$_{il}$ checks many fewer graphs in practice. For example, consider the $\mathcal{G}^1$ in Figure 1, with corresponding $\mathcal{H}$ for $u = 3$. RASL$_{re}$ constructs (not counting pruned single edges) 28,661 graphs; RASL$_{ie}$ constructs only 249 graphs; and RASL$_{il}$ considers only 47. For $u = 2$, these numbers are 413, 44, and 7 respectively. Unsurprisingly, these differences in numbers of examined graphs translate directly into wall clock time differences (Figure 3).

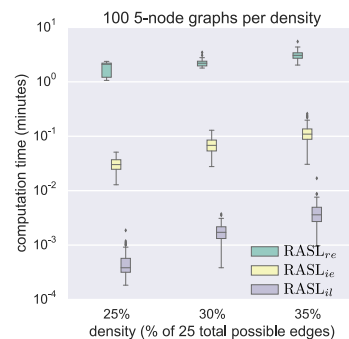

Figure 3: Run-time comparison.

## 4 Results

All three RASL algorithms take a measurement timescale graph $\mathcal{H}$ as input. They are therefore compatible with any structure learning algorithm that outputs a measurement timescale graph, whether Structural Vector Autoregression (SVAR) [11], direct Dynamic Bayes Net search [12], or modifications of standard causal structure learning algorithms such as PC [1, 13] and GES [14]. The problem of learning a measurement timescale graph is a very hard one, but is also not our primary focus here. Instead, we focus on the performance of the novel RASL algorithms.

First, we abstract away from learning measurement timescale structure and assume that the correct $\mathcal{H}$ is provided as input. For these simulated graphs, we focus on SCCs, which are the most scientifically interesting cases. For simplicity (and because within-SCC structure can be learned in parallel for a complex $\mathcal{H}$ [9]), we employ single-SCC graphs. To generate random SCCs, we (i) build a single simple loop over $n$ nodes, and (ii) uniformly sample from the other $n(n-1)$ possible edges until

we reach the specified density (i.e., proportion of the $n^2$ total possible edges). We employ density in order to measure graph complexity in an (approximately) $n$-independent way.

We can improve the runtime speed of RASL$_{\mathrm{re}}$ using memoization, though it is then memory-constrained for $n \geq 6$. Figure 3 provides the wall-clock running times for all three RASL algorithms applied to 100 random 5-node graphs at each of three densities. This graph substantiates our earlier claims that RASL$_{\mathrm{il}}$ is faster than RASL$_{\mathrm{ie}}$, which is faster than RASL$_{\mathrm{re}}$. In fact, each is at least an order of magnitude faster than the previous one.

RASL$_{\mathrm{re}}$ would take over a year on the most difficult problems, so we focus exclusively on RASL$_{\mathrm{il}}$. Unsurprisingly, run-time complexity of all RASL algorithms depends on the density of $\mathcal{H}$. For each of three density values (20%, 25%, 30%), we generated 100 random 6-node SCCs, which were then undersampled at rates 2, 3, and 4 before being provided as input to RASL$_{\mathrm{il}}$. Figure 4 summarizes wall clock computation time as a function of $\mathcal{H}$'s density, with different plots based on density of $\mathcal{G}^1$ and undersampling rate. We also show three examples of $\mathcal{H}$ with a range of computation runtime. Unsurprisingly, the most difficult $\mathcal{H}$ is quite dense; $\mathcal{H}$ with densities below 50% typically require less than one minute.

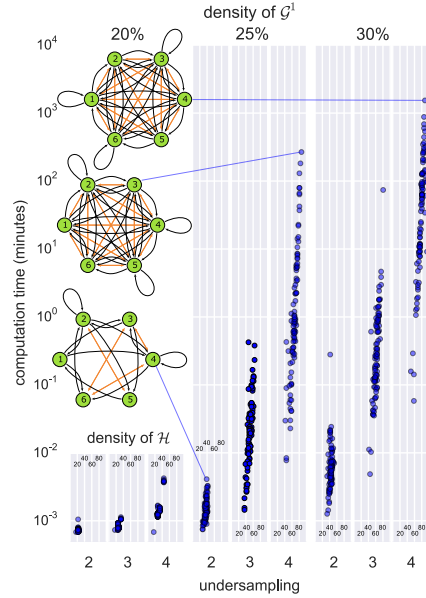

Figure 4: Run-time behavior.

## 4.1 Equivalence classes

We first use RASL$_{\mathrm{il}}$ to determine $[\![\mathcal{H}]\!]$ size and composition for varying $\mathcal{H}$; that is, we explore the degree of underdetermination produced by undersampling. The worst-case underdetermination occurs if $\mathcal{H}$ is a *super-clique* with every possible edge: $\forall X, Y[X \rightarrow Y \ \& \ X \leftrightarrow Y]$. Any SCC with $\gcd(\mathcal{L}_{\mathbf{S}}) = 1$ becomes a super-clique as $u \rightarrow \infty$ [9], so $[\![\mathcal{H}]\!]$ contains all such graphs for super-clique $\mathcal{H}$. We thus note when $\mathcal{H}$ is a super-clique, rather than computing the size of $[\![\mathcal{H}]\!]$.

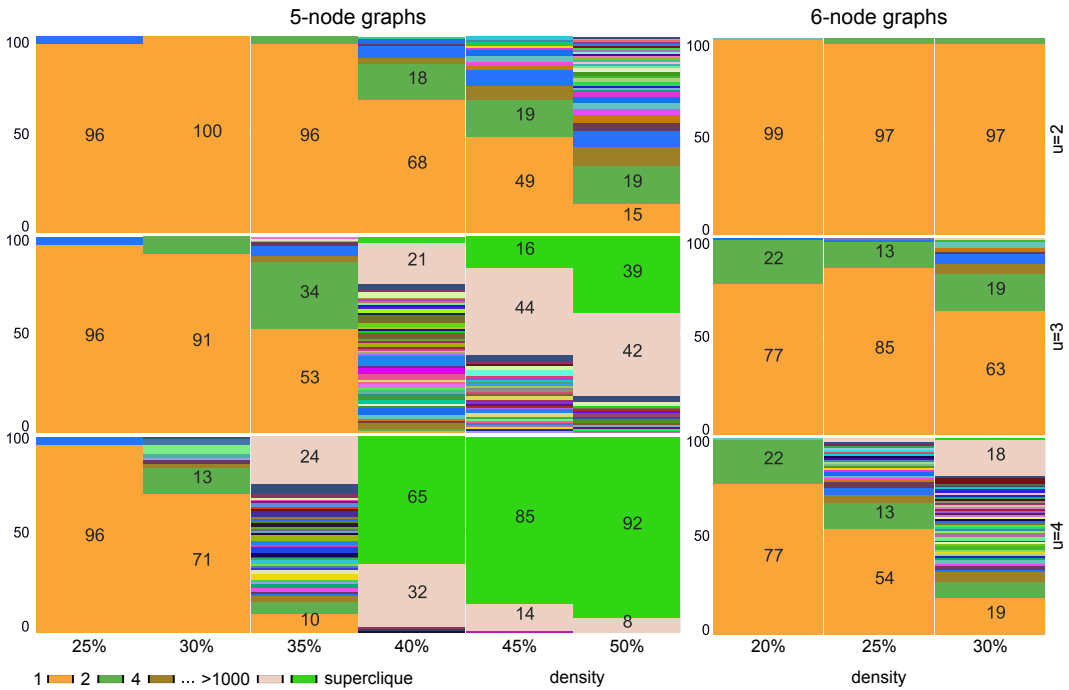

Figure 5: Size of equivalence classes for 100 random SCCs at each density and $u \in \{2, 3, 4\}$.

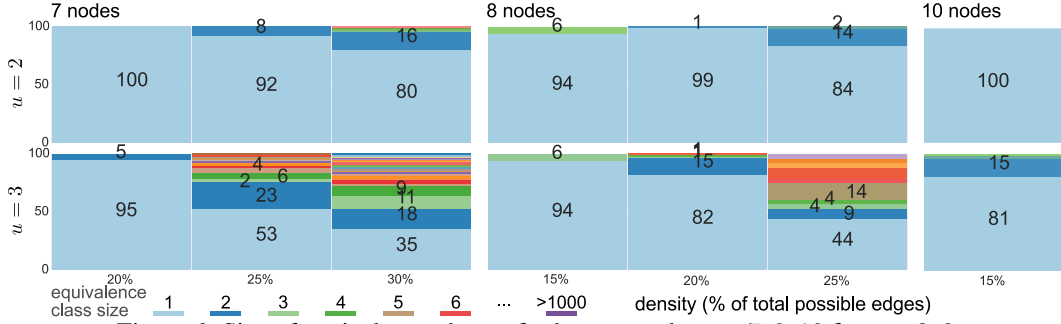

Figure 6: Size of equivalence classes for larger graphs $n \in 7, 8, 10$ for $u \in 2, 3$

Figures 5 and 6 plot equivalence class size as a function of both $\mathcal{G}^1$ density and the true under-sampling rate. For each $n$ and density, we (i) generated 100 random $\mathcal{G}^1$; (ii) undersampled each at indicated $u$; (iii) passed $\mathcal{G}^u = \mathcal{H}$ to RASL$_{il}$; and (iv) computed the size of $[\![\mathcal{H}]\!]$. Interestingly, $[\![\mathcal{H}]\!]$ is typically quite small, sometimes even a singleton. For example, 5-node graphs at $u = 2$ typically have singleton $[\![\mathcal{H}]\!]$ up to 40% $\mathcal{G}^1$ density. Even 10-node graphs often have a singleton $[\![\mathcal{H}]\!]$ (though with relatively sparse $\mathcal{G}^1$). Increased undersampling and density both clearly worsen underdetermination, but often not intractably so, particularly since even nonsingleton $[\![\mathcal{H}]\!]$ can be valuable if they permit post hoc inspection or analysis.

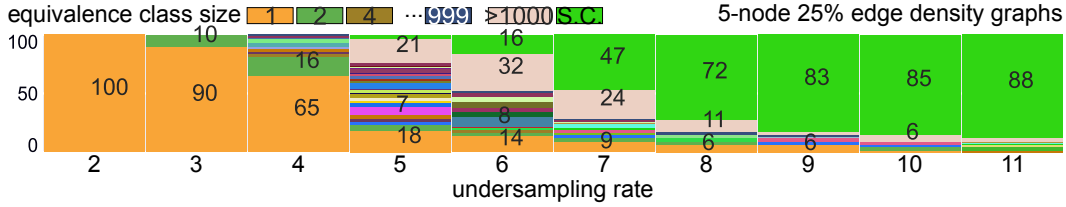

Figure 7: Effect of the undersampling rate on equivalence class size.

To focus on the impact of undersampling, we generated 100 random 5-node SCCs with 25% density, each of which was undersampled for $u \in \{2, \ldots, 11\}$. Figure 7 plots the size of $[\![\mathcal{H}]\!]$ as a function of $u$ for these graphs. For $u \leq 4$, singleton $[\![\mathcal{H}]\!]$ still dominate. Interestingly, even $u = 11$ still yields some non-superclique $\mathcal{H}$.

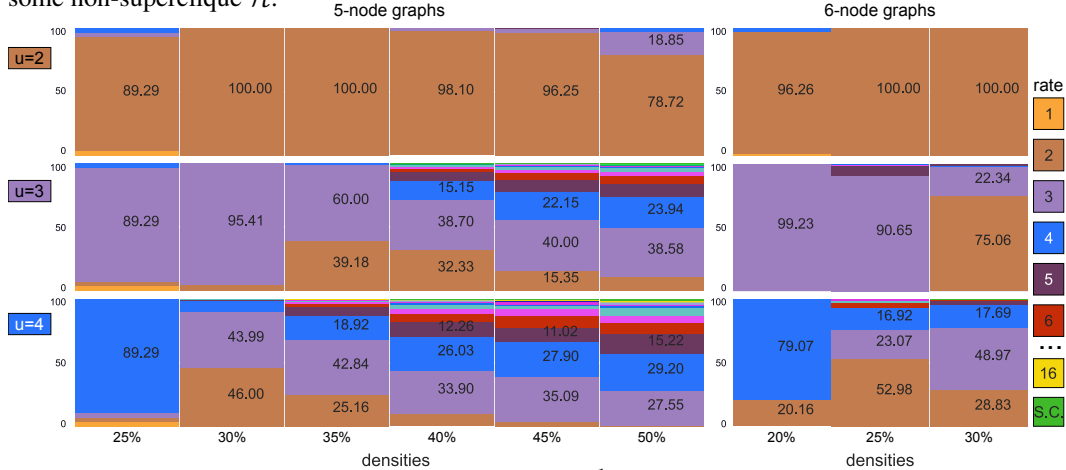

Figure 8: Distribution of $u$ for $\mathcal{G}^u = \mathcal{H}$ for $\mathcal{G}^1 \in [\![\mathcal{H}]\!]$ for 5- and 6-node graphs

Finally, $\mathcal{G}^1 \in [\![\mathcal{H}]\!]$ iff $\exists u [\mathcal{G}^u = \mathcal{H}]$, but the appropriate $u$ need not be the same for all members of $[\![\mathcal{H}]\!]$. Figure 8 plots the percentages of $u$-values appropriate for each $\mathcal{G}^1 \in [\![\mathcal{H}]\!]$, for the $\mathcal{H}$ from Figure 5. If actually $u_{true} = 2$, then almost all $\mathcal{G}^1 \in [\![\mathcal{H}]\!]$ are because of $\mathcal{G}^2$; there are rarely $\mathcal{G}^1 \in [\![\mathcal{H}]\!]$ due to $u > 2$. If actually $u_{true} > 2$, though, then many $\mathcal{G}^1 \in [\![\mathcal{H}]\!]$ are due to $\mathcal{G}^u$ where $u \neq u_{true}$. As density and $u_{true}$ increase, there is increased underdetermination in both $\mathcal{G}^1$ and $u$.

## 4.2 Synthetic data

In practice, we typically must learn $\mathcal{H}$ structure from finite sample data. As noted earlier, there are many algorithms for learning $\mathcal{H}$, as it is a measurement timescale structure (though small modifications are required to learn bidirected edges). In pilot testing, we found that structural vector autoregressive (SVAR) model [11] optimization provided the most accurate and stable solutions for $\mathcal{H}$ for our simulation regime. We thus employ the SVAR procedure here, though we note that other measurement timescale learning algorithms might work better in different domains.

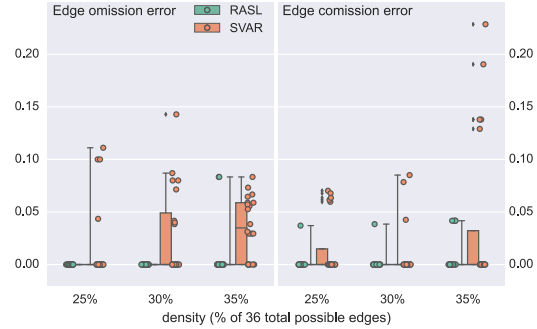

Figure 9: The estimation and search errors on synthetic data: 6-node graphs, $u = 2$, 20 per density.

To test the two-step procedure—SVAR learning passed to RASL$_{il}$—we generated 20 random 6-node SCCs for each density in $\{25\%, 30\%, 35\%\}$. For each random graph, we generated a random transition matrix $\mathbf{A}$ by sampling weights for the non-zero elements of the adjacency matrix, and controlling system stability (by keeping the maximal eigenvalue at or below 1). We then generated time series data using a vector auto-regressive (VAR) model [11] with $\mathbf{A}$ and random noise ($\sigma = 1$). To simulate undersampling, datapoints were removed to yield $u = 2$. SVAR optimization on the resulting time series yielded a candidate $\mathcal{H}$ that was passed to RASL$_{il}$ to obtain $[\![\mathcal{H}]\!]$.

The space of possible $\mathcal{H}$ is a factor of $\binom{n}{2}$ larger than the space of possible $\mathcal{G}^1$, and so SVAR optimization can return an $\mathcal{H}$ such that $[\![\mathcal{H}]\!] = \emptyset$. If RASL$_{il}$ returns $\emptyset$, then we rerun it on all $\mathcal{H}^*$ that result from a single edge addition or deletion on $\mathcal{H}$. If RASL$_{il}$ returns $\emptyset$ for all of those graphs, then we consider the $\mathcal{H}^*$ that result from two changes to $\mathcal{H}$, then three changes. This search through the 3-step Hamming neighborhood of $\mathcal{H}$ essentially always finds an $\mathcal{H}^*$ with $[\![\mathcal{H}^*]\!] \neq \emptyset$.

Figure 9 shows the results of the two-step process, where algorithm output is evaluated by two error-types: **omission error:** the number of omitted edges normalized to the total number of edges in the ground truth; **comission error:** number of edges not present in the ground truth normalized to the total possible edges minus the number of those present in the ground truth. We also plot the estimation errors of SVAR (on the undersampled data) to capture the dependence of RASL$_{il}$ estimation errors on estimation errors for $\mathcal{H}$. Interestingly, RASL$_{il}$ does not significantly increase the error rates over those produced by the SVAR estimation. In fact, we find the contrary (similarly to [6]): the requirement to use an $\mathcal{H}$ that could be generated by some undersampled $\mathcal{G}^1$ functions as a regularization constraint that corrects for some SVAR estimation errors.

## 5 Conclusion

Time series data are widespread in many scientific domains, but if the measurement and system timescales differ, then we can make significant causal inference errors [9, 15]. Despite this potential for numerous errors, there have been only limited attempts to address this problem [6, 7], and even those methods required strong assumptions about the undersample rate.

We here provided the first causal inference algorithms that can reliably learn causal structure from time series data when the system and measurement timescales diverge to an unknown degree. The RASL algorithms are complex, but not restricted to toy problems. We also showed that underdetermination of $\mathcal{G}^1$ is sometimes minimal, given the right methods. $[\![\mathcal{H}]\!]$ was often small; substantial system timescale causal structure could be learned from undersampled measurement timescale data. Significant open problems remain, such as more efficient methods when $\mathcal{H}$ has $[\![\mathcal{H}]\!] = \emptyset$. This paper has, however, expanded our causal inference "toolbox" to include cases of unknown undersampling.

**Acknowledgments**

SP & DD contributed equally. This work was supported by awards NIH R01EB005846 (SP); NSF IIS-1318759 (SP); NSF IIS-1318815 (DD); & NIH U54HG008540 (DD) (from the National Human Genome Research Institute through funds provided by the trans-NIH Big Data to Knowledge (BD2K) initiative). The content is solely the responsibility of the authors and does not necessarily represent the official views of the National Institutes of Health.

## Footnotes

[1]We use difference equations in our analyses. The results and algorithms will be applicable to systems of differential equations to the extent that they can be approximated by a system of difference equations.

[2]More precisely, we assume a dynamic variant of the Causal Sufficiency assumption, though it is more complicated than just "no unmeasured common causes."

[3]For set $\mathbf{B}$ of positive integers with $\gcd(\mathbf{B}) = 1$, $n_F$ is the max integer with $n_F \neq \sum_{i=1}^b \alpha_i B_i$ for $\alpha_i \geq 0$.

[4]This check requires at most $min(e_u, e_\mathcal{H}) + 1$ (fast) operations, where $e_u, e_\mathcal{H}$ are the number of edges in $\mathcal{G}^u, \mathcal{H}$, respectively. This equality check occurs relatively rarely, since $\mathcal{G}^u$ and $\mathcal{H}$ must be non-conflicting.

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
