[Supplementary Material · supp.pdf]

# Supplemental Materials: Rate-Agnostic (Causal) Structure Learning

**Sergey Plis**
The Mind Research Network,
Albuquerque, NM
s.m.plis@gmail.com

**David Danks**
Carnegie-Mellon University
Pittsburgh, PA
ddanks@cmu.edu

**Cynthia Freeman**
The Mind Research Network,
CS Dept., University of New Mexico
Albuquerque, NM
cynthiaw2004@gmail.com

**Vince Calhoun**
The Mind Research Network
ECE Dept., University of New Mexico
Albuquerque, NM
vcalhoun@mrn.org

## Abstract

Supplemental materials for Rate-Agnostic (Causal) Structure Learning.

## 1  Convergence

**Definition** Frobenius number is the largest integer unrepresentable by an integer weighted combination of a set of integers (denoted here as $n_F$).

**Definition** Graph diameter the length of the "longest shortest path" between any two graph nodes.

**Definition** Transient number is the length of the "longest shortest path" from a node that touches all simple loops of the SCC. We denote this path as $\tau$ and $length(\tau) = \gamma$.

In other words, $\gamma$ can be defined operationally as follows (see Figure 1):

1. for each node in the SCC find the shortest path that goes through the SCC and touches enough simple loops to make their gcd=1 and their Frobenius number $\leq n_F$ of the whole SCC (passing through just one node of a simple loop is enough);
2. choose the path of the maximum length out of these $n$ (number of nodes) shortest paths;
3. the length of this path is $\gamma$

**Theorem 3.1.** *If* $\gcd(\mathcal{L}_\mathbf{S}) = 1$*, then stabilization occurs at* $f \leq n_F + \gamma + d + 1$.[1]

*Proof.* Consider arbitrary nodes $X, Y$. Let $\tau_X$ be the "transient path" for $X$. By definition of Frobenius number and the fact that $length(\tau_X) \leq length(\tau) = \gamma$, we have that $\forall l \geq (n_F + \gamma)\exists \pi[\pi : X \to \ldots \to X \,\&\, length(\pi) = l]$. Let $\sigma$ be the shortest path from $X$ to $Y$. Clearly, $length(\sigma) \leq d$. Thus, $\forall l \geq (n_F + \gamma + d)\exists \rho[\rho = \pi \circ \sigma \,\&\, length(\rho) = l]$. Thus, for those $l$, there is a path from arbitrary $X$ to arbitrary $Y$. Because bidirected edges appear only for $u > k$ where $k$ is the length of the balanced paths, we must add 1 to ensure full convergence. $\square$

**Theorem 3.2.** *If* $\mathcal{G}^u = \mathcal{G}^v$ *for* $u > v$*, then* $\forall w > u \exists k_w < u[\mathcal{G}^w = \mathcal{G}^{k_w}]$.

$\gamma = 2$

length of the shortest path that starts on this node and touches both loops

$\gamma = 4$

length of the shortest path that starts on this node and touches both loops

| node | value |
|---|---|
| 1 | 2 |
| 2 | 1 |
| 3 | 0 |
| 4 | 1 |
| 5 | 0 |

| node | value |
|---|---|
| 1 | 4 |
| 2 | 3 |
| 3 | 2 |
| 4 | 1 |
| 5 | 0 |

F = 7
d = 4
converges in 13 steps at u=14

F = -1
d = 4
converges in 7 steps at u=8

maximum marked in red

Figure 1: Definition of $\gamma$ and computation of convergence undersampling rate for two SCCs with $\gcd(\cdot) = 1$

*Proof.* Let $\mathrm{ch}_{\mathcal{G}^u}(V_i)$ denote the children of $V_i$ in $\mathcal{G}^u$. In general, $\mathrm{ch}_{\mathcal{G}^{u+1}}(V_i) = \mathrm{ch}_{\mathcal{G}^1}(\mathrm{ch}_{\mathcal{G}^u}(V_i))$. Therefore, $\forall V_i[\mathrm{ch}_{\mathcal{G}^u}(V_i) = \mathrm{ch}_{\mathcal{G}^v}(V_i)] \Rightarrow \forall V_i[\mathrm{ch}_{\mathcal{G}^{u+1}}(V_i) = \mathrm{ch}_{\mathcal{G}^{v+1}}(V_i)]$. For $w = u + r$, repeated application of this rule (and using $\mathcal{G}^u = \mathcal{G}^v$ to "reset" the right-hand index) yields: $\forall V_i[\mathrm{ch}_{\mathcal{G}^w}(V_i) = \mathrm{ch}_{\mathcal{G}^{k_w}}(V_i)]$ for $k_w = v + r \bmod (u - v) < u$.

Since bidirected edges do not disappear as $u$ increases (and $\mathcal{G}^u = \mathcal{G}^v$), the bidirected edges in $\mathcal{G}^{k_w}$ are the same as those in $\mathcal{G}^u$. Suppose $V_i \leftrightarrow V_j$ is in $\mathcal{G}^w$ but not $\mathcal{G}^u$. Such an edge implies there is a balanced trek composed of directed paths $\pi_1, \pi_2$ with shared start $H$ of length $k \geq u$ in $\mathcal{G}^1$ and no such balanced trek of length $l < u$. This implies $H \to V_i$ and $H \to V_j$ both in $\mathcal{G}^k$, but not both in any $\mathcal{G}^l$. But this contradicts the conclusion of the previous paragraph that, for $k \geq u$, there exists $l < u$ such that $\forall V_i[\mathrm{ch}_{\mathcal{G}^k}(V_i) = \mathrm{ch}_{\mathcal{G}^l}(V_i)]$ (and so $\mathrm{ch}_{\mathcal{G}^k}(H) = \mathrm{ch}_{\mathcal{G}^l}(H)$). □

**Theorem 3.3.** *If $\mathcal{G}^1 \subseteq \mathcal{J}^1$, then $\forall u[\mathcal{G}^u \subseteq \mathcal{J}^u]$.*

*Proof.* It suffices to prove that, for arbitrary $u$, every directed and bidirected edge in $\mathcal{G}^u$ is contained in $\mathcal{J}^u$. Consider arbitrary $X \to Y$ in $\mathcal{G}^u$. This edge corresponds to some directed path $\pi$ in $\mathcal{G}^1$ from $X$ to $Y$ where $length(\pi) = u$. Since $\mathcal{G}^1 \subseteq \mathcal{J}^1$, every edge in $\pi$ is also in $\mathcal{J}^1$. Thus, $X \to Y$ must also be in $\mathcal{J}^u$. Similarly, arbitrary $X \leftrightarrow Y$ in $\mathcal{G}^u$ corresponds to a balanced trek $(\pi_1, \pi_2)$ in $\mathcal{G}^1$ with $X$ and $Y$ as the termini of each path, and where $length(\pi_1) = length(\pi_2) = k < u$. Every edge in $\pi_1$ or $\pi_2$ must also be in $\mathcal{J}^1$, so there is a suitable balanced trek in $\mathcal{J}^1$, so $X \leftrightarrow Y$ in $\mathcal{J}^u$. □

**Theorem 3.6.** *The RASL$_{re}$ algorithm is correct and complete.*

*Proof.* Correctness follows from the fact that we only add $\mathcal{G}^1$ to $[\![\mathcal{H}]\!]$ if we explicitly find $u$ such that $\mathcal{G}^u = \mathcal{H}$. For completeness, note that RASL$_{re}$ does not stop descending along a branch, even when it finds $\mathcal{G}^1 \in \mathcal{H}$. Thus, the only graphs that are not explicitly checked are those that are super-graphs of some $\mathcal{G}^1$ such that $\forall u[\mathcal{G}^u \not\subseteq \mathcal{H}]$. By Corollary 3.5, though, those graphs cannot possibly have $u$ such that $\mathcal{G}^u = \mathcal{H}$. Hence, every graph that could possibly be in $[\![\mathcal{H}]\!]$ is checked, and so RASL$_{re}$ is complete. □

**Lemma 3.7.** *If $u > 1$, then $\forall V \not\to W \in \mathcal{H}$, $\mathcal{G}^1$ cannot contain any of the following paths:*
*1. $\overset{\curvearrowright}{V} \to W$; 2. $\overset{\curvearrowright}{V} \to X \to W$; 3. $V \to \overset{\curvearrowright}{X} \to W$; 4. $V \to X \to \overset{\curvearrowright}{W}$; 5. $V \to \overset{\curvearrowright}{W}$.*

*Proof.* Proof by contradiction. If $\mathcal{G}^1$ contains a path of type (1), then $\forall u > 1$, there is a path $\pi : V \to \ldots \to V \to W$ with $u-1$ instances of $V$. $length(\pi) = u$, and so $\forall u > 1$, $V \to W \in \mathcal{G}^u$, so $\mathcal{G}^1 \notin [\![\mathcal{H}]\!]$. Similarly, if $\mathcal{G}^1$ contains a path of types (2) through (5), then $\forall u > 1$, there is a length-$u$ path $\pi : V \to \ldots \to W$ in $\mathcal{G}^1$, and so $\forall u > 1$, $V \to W \in \mathcal{G}^u$, so $\mathcal{G}^1 \notin [\![\mathcal{H}]\!]$. □

Figure 4: Enlarged version of Figure 4 in the main text.

**Lemma 3.8.** *If $u > 1$, then $\forall V \not\leftrightarrow W \in \mathcal{H} \; \nexists T[V \leftarrow T \rightarrow W] \in \mathcal{G}^1$*

*Proof.* Proof follows immediately from persistence of bidirected edges in undersampling.  □

## Footnotes

[1]All proofs are found in the supplement for clarity of exposition