[Reviews · NeurIPS 2015]

Submitted by Assigned_Reviewer_1

After explaining how to build a causal graphical model for down-sampled data by unrolling and marginalizing the original graphical model, finally leading to a compressed graph, the authors derive several results to retrieve the equivalence class of plausible original graphical model leading to this compressed graph. The first results (Theorems 3.1-3.3) help reduce the search space, while section 3.2 exploits these results to provide an algorithm to retrieve the equivalence class, and prove this algorithm is correct (Theorem 3.6). The performance of the algorithm is illustrated on synthetic data using Vector Autoregressive models.

Quality

The authors derive rigorously several theoretical results and an algorithm to solve the problem. A key publication on the topic is missing in the introduction: Discovering Temporal Causal Relations from Subsampled Data, Mingming Gong, Kun Zhang, Bernhard Schoelkopf, Dacheng Tao, Philipp Geiger, Proceedings of The 32nd International Conference on Machine Learning, pp. 1898-1906, 2015

Clarity

The manuscript is well written and organized.

Originality

As far as I know, these results are novel and of interest to the causal inference community.

Significance

While in the absence of extra assumptions, the equivalent classes generated by this approach might be too large to be exploited in applications, I think providing such results are important for further addressing the down-sampling problem.
Summary: The paper provides an interesting method to estimate a dynamic causal model in case measurements are a down-sampled with respect to the true time scale of the structural equations of the causal model. The method is well justified theoretically and illustrated on synthetic data.

Submitted by Assigned_Reviewer_2

The authors developed a nice set of theorems and corollaries that underpin the proposed algorithms for recovering causal structure in time series from downsampling in a rate-Agnostic setting. It is original in my opinion and has a potential to impact. My main concern is that the paper is done in a rush, which reduced its quality: (1) line 047: [insert citations] sounds like a senior member of the authors was instructing the 1st author, who missed the instruction. (2) typos and inconsistent notations: such as in line 130 where n_F ... should be included in the Theorem and more. (3) missing conclusion, an acknowledgements with no texts, and overall odd layout suggest the authors were running out of time. Overall, I like its originality a bit more and didn't spot any unfixable problem, thus I am leaning towards to accept it. During the discussion and rebuttal phase, I am open to change my rating upon whether a mature camera-ready version is likely to be produced.

After reviewers' discussion and seeing authors' response, my score remains unchanged.
Summary: This paper tries to answer a question: how to recover the original dynamic causal structure (or its equivalence class) from a much coarser causal structure obtained by undersampling with unknown rate (rate-Agnostic).

Submitted by Assigned_Reviewer_3

The paper introduces algorithms for finding the equivalence class of temporal DAGs, as defined by those compatible with a particular graph by a marginalization operation implied by measuring at coarser (and unknown) time resolutions. This is a hard problem. The paper is well-structured and presents three variants, including a simple but ineffective depth-first search algorithm.

I don't have much to comment on the methods themselves: they are clear enough, and I'm not sure how much more improvement one can really get. This seems to be inherently difficult, even more so than the problem of counting the number of models in the Markov equivalence class of a DAG, which as far as I know there is no tractable algorithm for doing so. As far as the experiments go, this is evident by the fact that only very small problems can be tackled.

I wonder if the authors are picking up the right fight. I understand the motivation, but some problems are just very hard. In DAG structure learning one is satisfied by just finding a representation of the equivalence class ("pattern") instead of the graphs compatible with it. Perhaps this is a more reasonable direction, although I'm not sure myself what a good non-trivial representation for this problem. Perhaps if one is really concerned about the time resolution of the measurements, it only makes sense to deal with continuous time processes, and the problem will need to be redefined.

As it is, this is a technically well-written paper coming from a clear direction, but it might be a tad too specialized even for a technical conference such as NIPS.

A short note on writing: a few definitions could have been made clearer, such as stabilization (not explicitly defined, but possible to understand), and Frobenius number (even the appendix that didn't clarify it for me: which weights are being used?).

UPDATE: I'm happy that the authors pointed out that scalability is not as bad as I first thought.
Summary: Algorithms for solving the problem of finding (causal) graphs compatible with a particular structure under a particular time resolution. A hard problem with a technically sound set of of solutions, but perhaps of very specialized interest only.

Submitted by Assigned_Reviewer_4

Strengths: - The algorithm proposed makes no assumptions about the rate of underlying system changes. - This paper introduces some theoretical results that describe important properties of the graphs in an equivalent class, such as theorem 3.2, 3.3 and lemmas 3.7 and 3.8.

Weaknesses: - The paper is rather well written but it has room for improvement. - The definition of trek in Section 2 is not clear and it seems that length(pi_2) is missing in the definition. - Based on the definition of G in section 2, the only edges are from V_i^(t-1) to V^t_j. This is not consistent with having a directed path from V_i^(t-u) to V_j^t if V_j has no influence on V_i. - In all the algorithms, it is necessary to check if for some u, G^u=H. what is the complexity of such an operation? - Discussion about the performance and scalability of the proposed algorithms would be helpful. It seems the algorithms are not very practical for large network as the simulations are only for graphs of size 5 or at most 6. - A proper definition for the stabilization is required. It is not clear what the goal of Theorem 3.1 is.
Summary: The authors study an interesting problem, learning the equivalent class of graphs that could explain the relationships between measurement data. These measurements occur at a slower rate than the underlying system changes which is unknown. The authors consider a Markov system of first order and suggest three algorithms for learning such equivalent class of graphs. Furthermore, they theoretically justify the correctness of their algorithms.

Submitted by Assigned_Reviewer_5

This paper proposes a framework and algorithms for learning causal structures from "undersampled" time series data, i.e. where the sampling occurs at a lower rate than the system, when it is unknown what the ratio of undersampling is; the only previous approach to deal with the undersampling problem required that the ratio of undersampling be small and known.

The authors attempt to learn an equivalence class of structures for which an undersampling ratio exists that is consistent with the observed time series by learning a structure for the time series and then iterating over possible structures for the system while pruning the solution tree. The authors prove that their first algorithm, RASL_{re} is correct and complete and then introduce more efficient refinements, RASL_{ie} and RASL_{il}. The authors devote a significant portion of the paper to simulation studies comparing these three algorithms in different contexts where the structure for observed time series is known and then the authors provide one synthetic data example when this structure is unknown.

The authors are dealing with a very difficult problem that is often shoved under the rug in related approaches. The framework for how they've approached this problem is very interesting and may turn out to be very fruitful, but the paper appears to be very preliminary at this point and is missing some things.

The results section focuses a lot of attention to comparing the 3 different versions of RASL when the correct structure for the undersampled time series is known, which is essentially an implementation issue, at the expense of testing the full procedure in a realistic context when this graph is unknown. Only one synthetic data example is given with a 5 node structure and the discussion is limited. I'm left not really knowing what the significance of the edge omission and commission rates are, whether they converge, or how scalable this algorithm. The lack of discussion is sort of a problem throughout the results section and exacerbated by the fact that there is no conclusion. There are a lot of interesting observations given, but not much in terms of holistic conclusions about the performance of the algorithm. Aside from the last two sentences in 4.2, the authors don't really do much to "sell" their procedure.

The authors also neglect the issue of learning the structure of the undersampled system. Several methods are mentioned, but the authors do not discuss the assumptions or implications of using one of these methods over another or whether they are correct or complete (which seems important since the authors prove their method which uses the results from these methods is correct and complete). Results are only given SVAR because the authors say it gave accurate and stable solutions in pilot testing, but they don't discuss further as to why they picked SVAR and whether they tested other methods. Including results for these other methods is another way that the results section could be improved.

The authors also mention a number of possible application areas at the beginning of the paper, but do not apply their method to any real data. Some attempt to assess the performance on real data (even if the absolute ground truth is unknown), would also improve the results section.
Summary: This paper proposes a framework and algorithms for learning causal structures from

undersampled time series data when the ratio of undersampling is unknown. The authors' approach is interesting, but very preliminary and lacks sufficient experimental results and discussions, particularly in regards to how to approach a key part of the problem.

Author Feedback
Author rebuttal: We thank the reviewers for their valuable feedback about our paper. It was gratifying that there was widespread agreement about the importance of this previously under-discussed problem.

Several reviewers (particularly Reviewer_7) noted the difficulty of the problem that we tackle, and expressed concerns about whether our aspirations are too high. These concerns seemed to be partially motivated by the limited graph sizes in our simulations. We agree that this is a very difficult problem, but we believe that this paper not only shows this difficulty, but also provides the first algorithms that solve it. As noted by Reviewer_4, previous algorithms for undersampled data assumed that the undersampling rate is known and small. To our knowledge, our paper provides the first algorithms that eliminate this assumption. We certainly do not think that the RASL algorithms are the last word or best possible methods, but they provide a key baseline from which to develop and test improved future algorithms. Moreover, RASL_{il} can already run on variable sets of size 10 and more, but it takes sufficiently long that it was impractical to run the thousands of structures required for a systematic simulation study.

Relatedly, Reviewer_4 expressed significant concern about the minimal discussion or simulations in which the measurement (timescale) graph also had to be learned from time series data. We apologize for not providing more details about our reasons for using SVAR estimation in that step. Our intention was not to hide anything, but rather to maintain our focus on the "measurement structure --> causal structure" step of the overall process, for (at least) two reasons. First, in our view, this latter step is the theoretically novel one; there are multiple algorithms for learning measurement structures from data (e.g., SVAR estimation, Granger causality, dynamic Bayes net structure learning algorithms, modifications of i.i.d. structure learning algorithms, etc.), each with their own strengths and weaknesses. In contrast, there are (to our knowledge) no other algorithms that infer causal structure from measurement structure without knowledge of the measurement/causal timescale ratio. We thus thought it important that the RASL algorithms be the primary focus of our presentation and testing.

Second, we believe it is a significant strength of the RASL procedures that they are agnostic about the source of the input graph. As noted above, different algorithms for learning measurement structures have different performance profiles, and RASL allows researchers to use the best such algorithm for their particular problem domain. In our simulations (both in this paper and others), SVAR estimation has worked best, but we recognize that this could be due to idiosyncratic features of our simulation procedures. We certainly do not claim that SVAR estimation is always the best way to learn the measurement structure. Relatedly, scientists may have significant domain knowledge that enables them to manually provide a measurement timescale structure as input without deriving it from any single dataset (e.g., by aggregating results from multiple studies). In those contexts, the problem of learning measurement structure is further separable from RASL's performance. We absolutely agree with Reviewer_4 that the problem of learning measurement structure is a hard and important one. In this particular paper, however, we tried to maintain our focus on the novel, complementary challenge of taking those measurement structures (however they were obtained) and learning causal timescale structures that could have produced them.

Reviewer_3 asks about the complexity of checking whether G^u = H. In general, this query requires at most min(e_u, e_H)+1 operations, where e_u & e_H are the number of edges in G^u & H, respectively. Each operation is very fast (since it is just a comparison of two bits). Moreover, this equality check occurs relatively rarely, since it arises only if G^u and H are already known to be non-conflicting.

Finally, multiple reviewers expressed concerns about elements of the presentation. As Reviewer_2 suspected, there were unexpected events that led to the submitted paper having some unfortunate errors and omissions due solely to time constraints. These would all certainly be fixed in the final version, and we think that the paper would be stronger as a result, especially with a Conclusion that clearly states the results of our research. (Thanks also to Reviewer_1 for the reference to an important related paper; we learned of it only after we had submitted the present manuscript.)